# Corneal Complications after COVID-19 Vaccination: A Systemic Review

**DOI:** 10.3390/jcm11226828

**Published:** 2022-11-18

**Authors:** Li-Ying Huang, Chun-Chi Chiang, You-Ling Li, Hung-Yin Lai, Yi-Ching Hsieh, Ying-Hsuen Wu, Yi-Yu Tsai

**Affiliations:** 1Department of Ophthalmology, China Medical University Hospital, China Medical University, Taichung 404327, Taiwan; 2School of Medicine, College of Medicine, China Medical University, Taichung 406040, Taiwan; 3Department of Optometry, Asia University, Taichung 413305, Taiwan

**Keywords:** coronavirus, COVID-19, SARS-CoV-2, vaccine, eye, cornea

## Abstract

Multiple vaccines are now being used across the world, and several studies have described cases of corneal graft rejection following the administration of the COVID-19 vaccine. The purpose of this article is to review the corneal adverse event that occurred following COVID-19 vaccine administration. The literature search was conducted in March 2022 using MEDLINE, PubMed, and the Cochrane Database of Systematic Reviews. A total of 27 articles, including 37 cases, have documented corneal adverse events that occurred following COVID-19 vaccination. The mean age was 60 ± 14.9 years (range, 27–83 years). The most common events were acute corneal graft rejection (*n* = 21, 56.8%), followed by herpes zoster ophthalmicus (*n* = 11, 29.7%) and herpes simplex keratitis (*n* = 2, 5.4%). The mean time from vaccination to the event was 10 ± 8.5 days (range, 1–42 days) after the first or second dose of vaccine. All patients with corneal graft rejection, immune-mediated keratolysis, and peripheral ulcerative keratitis (PUK) (*n* = 24, 64.9%) were managed topically with or without oral corticosteroids. Patients with herpes zoster ophthalmicus and herpes simplex keratitis were managed with oral antiviral agents. Two patients received penetrating keratoplasty due to keratolysis after invalid topical treatment. Disease resolution was noted in 29 patients (78.3%), whereas 3 (8.1%) had persistent corneal edema after graft rejection, 1 (2.7%) had corneal infiltration after HZO, and 4 (10.8%) were not mentioned in the articles. Corneal adverse events could occur after COVID-19 vaccination. After timely treatment with steroids or antiviral agents, most of the events were mild and had a good visual outcome. Administrating or increasing steroids before vaccination may be useful for the prevention of corneal graft rejection. However, the prophylactic use of antiviral treatments in patients with a herpes viral infection history is not recommend.

## 1. Introduction

The outbreak of the coronavirus disease 2019 (COVID-19) was caused by the severe acute respiratory syndrome coronavirus 2 (SARS-CoV-2) in December 2019. Globally, there are four types of COVID-19 vaccines available, including the recombinant messenger RNA (mRNA) vaccines (Pfizer/BioNTech BNT162b2 and Moderna mRNA1273); the protein subunit vaccines (Novavax); the adenovirus vector-based vaccines (Oxford–AstraZeneca ChAdOx1 nCoV-19 and Janssen Johnson & Johnson Ad26.COV2.S); and the inactivated virus vaccines (Sinovac, Sinopharm, and Covaxin).

Wang et al., Haseeb et al., and Lee et al. described ocular complications following vaccination for COVID-19 that may appear on the eyelid, orbit, cornea and ocular surface, retina, uvea, nerve, and vessel, as well as ocular motility disorders [1,2,3]. The most common ocular adverse events present after COVID-19 vaccination are optic neuritis and uveitis, followed by herpes zoster ophthalmicus and ischemic optic neuropathy [1]. The possible mechanisms of adverse events after COVID-19 vaccination have been reported on in numerous reports and retrospective case studies. The purpose of this literature review is to provide an overview of COVID-19 vaccine-associated corneal adverse events.

## 2. Materials and Methods

The literature search was conducted on 23 March 2022 using three databases, including EMBASE, PubMed, and the Cochrane Database of Systematic Reviews. We used the following key words for searching: ((“2019-nCov” OR “nCoV-19” OR “SARS-CoV-2” OR “SARS-CoV2” OR “SARSCoV2” OR “SARS2” OR “COVID” OR “coronavirus” OR “coronavirus disease” OR “severe acute respiratory syndrome”) vaccine) AND “cornea”. No time range limits were set for the literature search. Articles were included if they were case reports or retrospective studies describing any adverse corneal manifestations following all vaccinations against COVID-19. A flow chart for article selection is shown in Figure 1. A total of 64 articles were found after searching Embase, PubMed, and Cochrane according to the search strategy. Among these, 17 articles were excluded due to duplicate articles, and 20 articles were excluded due to being irrelevant. Table 1 provides the summaries of all the included studies from the peer-reviewed literature.

## 3. Results and Discussion

In our review, a total of 27 reports (37 patients, 39 eyes) documented corneal adverse events that occurred following COVID-19 vaccination. Of these 37 patients, 17 (45.9%) were females, and 20 (54.1%) were males. The mean age at the time of presentation was 60 ± 14.9 years (range, 27–83 years). The average time from vaccination to the development of ocular symptoms was 10 ± 8.5 days (range, 1–42 days) after the first or second doses of vaccine. Of the 31 cases, BNT162b2 mRNA SARS-CoV-2 (BioNTech/Pfizer, Mainz, Germany) was reported 13 (35.1%) times, AZD1222 ChAdOx1 nCoV-19 (AstraZeneca, Cambridge, UK, also marketed as the COVISHIELD Serum Institute of India vaccine) was reported 3 (8.1%) and 6 (16.2%) times, Moderna COVID-19 Vaccine (ModernaTX, Inc., Cambridge, MA, USA) was reported 10 (27.0%) times, Janssen (Johnson & Johnson, Ad26.COV2.S) was reported 1 (2.7%) time, and Corona Vac (Sinovac Biotech Ltd., Beijing, China) was reported 3 (8.1%) times. Most of the cases (25, 67.6%) were after the first dose, and nine (24.3%) were after the second dose.

The most common event was acute corneal graft rejection (*n* = 21, 56.8%), followed by herpes zoster ophthalmicus (*n* = 11, 29.7%) and herpes simplex keratitis (*n* = 2, 5.4%). One patient had corneal epithelium rejection after conjunctival limbal graft surgery, one patient had keratolysis in both eyes, and another patient had unilateral PUK with nodular scleritis (Table 2).

### 3.1. Corneal Graft Rejection

There were 21 reported cases of acute corneal graft rejection, occurring from one day to six weeks following COVID-19 vaccine administration, according to the peer-reviewed literatures. The mean age at the time of presentation was 65.6 ± 11.7 years (range, 35–83 years). The adverse events developed at an average of 64.1 ± 100.7 months after surgery. The average interval between the vaccination and the rejection was 10.5 ± 8.9 days. Thirteen (68.4%) of the cases occurred in patients with penetrating keratoplasty (PKP), and three (10.5%) of the cases occurred in patients with Descemet stripping automated endothelial keratoplasty (DSAEK), while the five (21.1%) remaining patients had Descemet membrane endothelial keratoplasty (DMEK) [Table 3].

Seven of the cases with PKP were regrafts, with one patient having steroid-induced glaucoma following the first PKP graft for keratoconus; one patient had a previous failed DSAEK for pseudophakic bullous keratopathy (PBK), while the other patient had a previous failed PKP. One of the cases with DSAEK was regrafted due to previous failed graft for Fuchs endothelial corneal dystrophy (FECD). One of the DMEK cases was retransplanted due to herpes simplex keratitis-related graft failure for PBK. Fifteen (71.4%) of the 21 cases of graft rejection were reported following recombinant mRNA vaccine (Moderna/ BNT162b2) vaccine administration, and five (23.8%) of the 19 cases were following adenovirus vector vaccine (AZD1222/COVISHIELD) administration, while the remaining case (1, 4.8%) occurred after the inactivated virus vaccine (Sinovac) administration. Seven (33.3%) cases were reported following the second vaccine dose, while the other (14, 66.7%) graft rejection occurred after the first dose.

#### 3.1.1. Mechanisms

Corneal transplantation has a relatively low rate of graft rejection due to unique ocular immune privilege, lack of vasculature, expression of immune cell inhibitory cytokines preventing activation of T cells, and low expression of MHC molecules [27]. The most frequent cause of graft failure is allogeneic rejection. The graft rejection can occur at any cellular levels of the cornea, most commonly resulting in endothelial dysfunction, inducing irreversible loss of donor endothelial cells, with subsequent stromal edema and reduced corneal clarity [7].

Corneal graft rejection after vaccine administrations have been previously described following other types of immunization, such as influenza, hepatitis B, tetanus, yellow fever, and tetanus toxoid vaccination. However, the underlying pathophysiological mechanisms of vaccine-related corneal transplant rejections remain poorly understood [28,29,30]. The COVID-19 vaccines have different immunization mechanism, based on their antigenic design. Several mechanisms and the pathogenesis of corneal graft rejection have been proposed. Non-replicating viral vector vaccines, including AZD1222 and COVISHIELD, are encoded with a spike protein of SARS-CoV-2 on an adenovirus derived from chimpanzees. Therefore, adenoviral vector vaccines can cause ocular diseases by inducing immunologic responses to the spike antigen or to components of the chimpanzee, or the human adenovirus [4].

All vaccines introduce into the body a harmless piece of a particular bacteria or virus, triggering an immune response. Most vaccines contain weakened or dead bacteria or a virus. However, mRNA vaccines work by introducing a piece of mRNA that corresponds to a viral protein, usually a small piece of a protein found on the virus’s outer membrane. These mRNA SARS-CoV-2 (BioNTech/Pfizer, Mainz, Germany) vaccines use nucleoside-modified mRNA formulated in lipid particles, which enables the delivery of the nucleoside-modified mRNA into host cells to allow expression of the SARS-CoV-2 spike antigen. These mRNA vaccines may generate both adaptive humoral and cellular immune responses, providing adjuvant activity to drive dendritic cell maturation and thus enhancing T-cell and B-cell immune responses. During the process of generating antibodies, the nanoparticles activate dendritic cells and thus stimulate B-cell and T-cell immune responses, including high levels of neutralizing antibody titers, interferon gamma (IFNγ), and antigen-specific CD8+ and TH1-type CD4+ T-cell responses [7,31,32,33,34]. In particular, the IFNγ-producing Th1-type CD4+ lymphocytes have been the primary mediators of corneal allograft rejection [35].

Steinemann et al. hypothesized that increased vascular permeability following vaccination impairs the corneal immune privilege, and immunization may increase MHC class II complex antigen expressions of the cornea, which can induce donor cells without MHC expression to be targeted by host immune cells due to poor immunogenicity [29,36]. Shah et al. hypothesized that a mechanism of vaccination-induced rejection resulted from a disruption in the immunoregulatory state of the eye that normally confers immune privilege. Such an overall proinflammatory state postvaccination could enhance the body’s immune response to foreign antigens, such as transplanted corneal tissues [6].

#### 3.1.2. Treatment and Outcomes

Prompt initiation of intensive topical and/or systemic corticosteroid therapy resulted in favorable outcomes, and it resolved rejection events and prevented graft failure in 16 (76.1%) cases. For the 10 affected eyes that had reported visual acuity, the mean baseline visual acuity was logMAR 0.14, whereas mean visual acuity at presentation after vaccination was logMAR 0.71, and final visual acuity after treatment was logMAR 0.17. The average recovery time was 3.7 ± 2.6 weeks (range, 2 days to 8 weeks).

Three of the 18 cases persisted or progressed into corneal edema despite topical or intracameral steroid treatment. Yu et al. described the case of a 51-year-old male who presented eye pain, photophobia, and blurred vision 3 days after his first dose of the SARS-CoV-2 mRNA-1273 Moderna vaccine. Despite a pulse of topical steroids, the graft cornea tissue continued to fail, with progressively worsening clarity and reduction in visual acuity to hand motion (HM). A repeat corneal transplant will be arranged after full immunization and stability [7]. Balidis et al. also reported two cases of poor response to corticosteroid treatment. No substantial improvement was noticeable after treatment using corticosteroid drops (dexamethasone) and intracameral corticosteroid injection (fortecortin) for a patient who had repeated PK and another patient who had second Descemet stripping endothelial keratoplasty [4].

Although there is also no consensus opinion among corneal transplant surgeons, Lockington et al. investigated 142 corneal keratoplasty specialists, and 26.2% of whom initiated or increased topical steroids in the perivaccination period for their post-keratoplasty patients receiving vaccinations of any type [37]. Administration of local or oral steroids before the vaccine administration could be useful for preventions [30]. Close follow-ups after vaccination can allow the notice of rejection and prescribe timely treatment at an early stage.

### 3.2. Corneal Epithelial Rejection of LR-CLAL

Presa et al. reported on a 27-year-old patient who underwent living-related conjunctival limbal allograft (LR-CLAL) surgery 4 years and 7 months earlier and developed acute epithelial rejection 15 days after receiving his first dose of the recombinant mRNA vaccine (Moderna). There was no history of rejection. After topical and oral steroid treatment for one week, the rejection line and epitheliopathy decreased. Before the second dose of vaccination, topical and systemic immunosuppression was increased, and there were no further rejection episodes. Cheung and Eslani reported that acute rejection occurred in 30.2% of LR-CLAL eyes, and 21.2% of these patients progressed to develop a failed surface [38]. Higher risks of rejection were noted for inflammatory causes of limbal stem cell deficiency, including Steven–Johnson syndrome and mucous membrane pemphigoid.

### 3.3. Herpetic Eye Disease

Eleven cases of herpes zoster ophthalmicus, occurring within 1 day to 4 weeks following COVID-19 vaccine administration, were reported in the peer-reviewed literatures [18,19,20,21,22,23]. Seven cases were reported after receiving the first dose of the recombinant mRNA vaccine (Moderna/ BNT162b2), three cases occurred after receiving the adenovirus vector vaccine (COVISHIELD/Johnson & Johnson), and one case occurred after receiving the second dose of the inactivated virus vaccine (Sinovac). Ophthalmic involvement was limited to conjunctival injection in eight cases and corneal involvement with pseudodendritic lesion or corneal edema in three cases, while one of these patients had herpetic corneal endotheliitis. Resolutions were achieved in most cases with routine systemic antiviral treatment, except one patient, who had persistent corneal infiltration.

Reactivation of the herpes virus has been associated with increasing age, HIV infection, cancer, physical or emotional stress, fever, exposure to ultraviolet light, tissue damage, and immunosuppression [39]. Data from the Adverse Event Reporting System showed that the population prevalence of ocular herpes zoster after vaccination was 0.5 cases per million doses or less, and the incidence of ocular herpes simplex was 0.05 cases per million doses or less [1].

Herpes virus reactivation was reported following the administration of the influenza, hepatitis A, and rabies vaccinations. The exact mechanisms that trigger the reactivations of herpes viral infection after COVID-19 vaccination remain elusive, but vaccine-induced immunomodulation was reported to be the cause for virus reactivation [19,40]. Hypotheses of the mechanisms for herpes virus reactivation following vaccination include molecular mimicry, in which the host proteins are mimicked by those within the vaccine, triggering a host response [24]. A further proposed mechanism includes autoinflammation triggered by the vaccine, with a possible reduction in neurotrophin, allowing HSV replication. Further, a distraction of humoral response due to the vaccination may lead to the loss of the immunological control of HSV [41,42]. There is also no sufficient evidence to support prophylactic antiviral therapy in people with a history of herpetic keratitis [20].

### 3.4. Keratolysis

Khan et al. reported a 48-year-old male with immune-mediated pathology for keratolysis that occurred at 3 weeks after receiving the first dose of the recombinant vaccine (COVISHIELD). He was under the therapy of fortified topical antibiotics (vancomycin 5% and tobramycin 1.3% every 2 h), cycloplegic (homatropine hydrobromide 2% thrice a day), antiglaucoma medication (timolol maleate 0.5% twice a day and oral acetazolamide 250 mg twice a day), broad spectrum oral antibiotics (ciprofloxacin 500 mg twice a day), oral acyclovir 400 mg five times a day, and oral vitamin C 500 mg thrice a day. A systemic workup for bilateral stromal melting and necrosis showed an increased erythrocyte sedimentation rate (40 mm/h), but other examinations were within the normal limits. Microbiological investigations showed negative bacterial and fungal culture reports. The patient underwent sequential tectonic penetrating keratoplasty of both eyes, due to corneal perforation. The long-term outcome after surgery was not documented.

Any other causes of bilateral keratolysis and the presence of lymphocytes, macrophages, and eosinophils in the host cornea were excluded. Khan et al. suggested that an intense immunogenic and hypersensitivity response was triggered by the ChAdOx1 nCoV-19 vaccine (COVISHIELD, Serum Institute of India/Oxford AstraZeneca), but the underlying mechanisms are still unclear [25].

### 3.5. Peripheral Ulcerative Keratitis (PUK)

Penbe et al. reported a 67-year-old patient with unilateral progressive vision-threatening PUK with nodular scleritis after receiving the inactive vaccine for COVID-19 [26]. The extended workup for autoimmune and infectious etiologies for PUK all returned negative. The patient was treated with oral steroids, azathioprine, topical cyclosporine, topical dexamethasone, and corneal amniotic membrane grafting four times and then finally underwent penetrant keratoplasty.

This was the first case report of vaccine-related PUK, and no other vaccine had been reported to be related to PUK. The mechanisms and relationships are unclear, but vaccine-induced immunomodulation may be the cause.

## 4. Conclusions

We reported the largest review of corneal adverse events that occurred following COVID-19 vaccination. The most common corneal adverse event is corneal graft rejection. Most of the rejection occurs within the proximity of 1–2 weeks after vaccination. Prompt topical or systemic corticosteroid therapy yields good results. In higher-risk situations, intravenous corticosteroids may be of benefit. Although rare ocular manifestations may develop after vaccination, people are still encouraged to get vaccinated, since the benefits outweigh the risks. Administrating steroids or increasing steroid dosage before vaccine administration could be useful for graft rejection preventions, especially in high-risk patients. Due to the low incidence of herpetic keratitis and the lack of significant evidence, we do not suggest prophylactic antiviral therapy in patients with a history of herpetic keratitis.

## Figures and Tables

**Figure 1 jcm-11-06828-f001:**
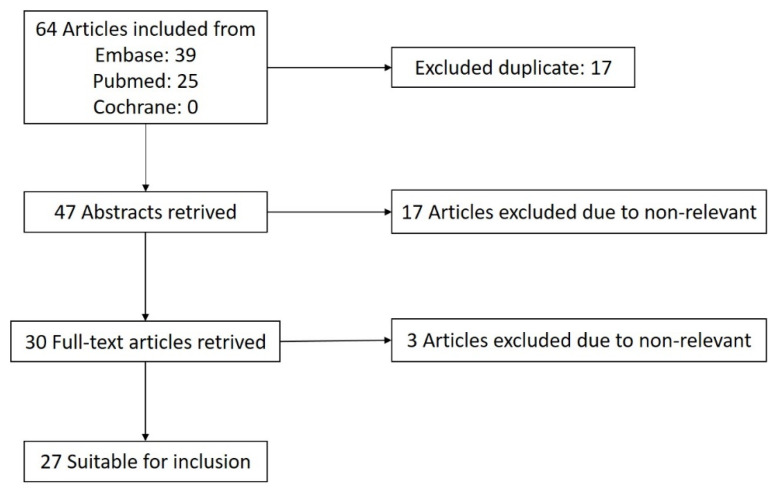
Flow chart of article selection.

**Table 1 jcm-11-06828-t001:** Review of literatures about corneal complications after COVID-19 vaccination.

Study	Age	Sex	Vaccine	Dose	IVR (Days)	Diagnosis	Treatment	Outcome
Balidis M et al. 2021 [4]	77	F	Moderna	1	7	Corneal graft rejection	SCI, Topical, IV steroid	Resolution
	64	F	Moderna	2	7	Corneal graft rejection	Topical, ICI steroid	Persisted corneal edema
	69	M	AZD1222	1	5	Corneal graft rejection	SCI, oral, topical steroid	Resolution
	63	M	AZD1222	1	10	Corneal graft rejection	Topical steroid, hypertonic ointment	Persisted corneal edema
Parmar DP et al. 2021 [5]	35	M	COVISHIELD	1	2	Corneal graft rejection	Topical steroid Atropine, IV methylprednisolone	Resolution
Shah AP et al. 2022 [6]	74	M	Moderna	1	7	Corneal graft rejection	Topical steroid	Resolution
	61	F	Moderna	2	7	Corneal graft rejection	Topical steroid	Resolution
	69	F	Moderna	2	14	Corneal graft rejection	Topical steroid	Resolution
	77	M	Moderna	2	7	Corneal graft rejection	Topical steroid	Resolution
Yu S et al. 2022 [7]	51	M	Moderna	1	3	Corneal graft rejection	Topical steroid	Progression to HM
Phylactou M et al. 2021 [8]	66	F	BNT162b2	1	7	Corneal graft rejection	Topical steroid	Resolution
	83	F	BNT162b2	2	21	Corneal graft rejection	Topical steroid	Resolution
Ravichandran S et al. 2021 [9]	62	M	COVISHIELD	1	16	Corneal graft rejection	Not mentioned	NA
Simão MF et al. 2022 [10]	63	F	Sinovac	1	1	Corneal graft rejection	Topical steroid, timolol	Resolution
Wasser LM et al. 2021 [11]	73	M	BNT162b2	1	13	Corneal graft rejection	Topical, oral steroid	Resolution
	56	M	BNT162b2	1	14	Corneal graft rejection	Topical, oral steroid	Resolution
Rallis KI et al. 2021 [12]	68	F	BNT162b2	1	3	Corneal graft rejection	Topical steroid, oral acyclovir	Resolution
Nioi M et al. 2021 [13]	44	F	BNT162b2	1	13	Corneal graft rejection	Topical steroid, vitamin D	Resolution
Rajagopal R et al. 2022 [14]	79	M	COVISHIELD	2	42	Corneal graft rejection	Topical, oral steroid	Resolution
Abousy M et al. 2021 [15]	73	F	BNT162b2	2	14	Corneal graft rejection	Topical steroid	NA
Crnej et al. 2021 [16]	71	M	BNT162b2	1	7	Corneal graft rejection	Topical steroid, oral acyclovir	Resolution
de la Presa M et al. 2022 [17]	27	F	Moderna	1	15	Corneal epithelial rejection of LR-CLAL	Topical, oral steroid, Mycophenolate mofetil	Resolution
Lazzaro DR et al. 2022 [18]	46	M	BNT162b2	1	1	HZO with corneal involvement	Oral valacyclovir, topical ganciclovir	Persistent corneal infiltrates
	34	F	BNT162b2	1	14	HZO	Antivirals	Resolution
Rehman O et al. 2022 [19]	35	M	COVISHIELD	1	3	HZO	Oral valacyclovir, topical acyclovir, moxifloxacin	Resolution
	40	M	COVISHIELD	1	28	HZO	Oral valacyclovir, topical moxifloxacin	Resolution
Li S et al. 2021 [20]	51	M	Sinovac	2	2	HZO	Topical steroid, topical and oral ganciclovir	Resolution
Papasavvas I et al. 2021 [21]	73	F	BNT162b2	NA	16	HZO	Oral valacyclovir	Resolution
	69	F	BNT162b2	1	10	HZO	Oral valacyclovir, topical acyclovir	Resolution
	72	F	Moderna	1	13	HZO	Oral valacyclovir, topical acyclovir, antibiotic, steroid	Resolution
Furer et al. 2021 [22]	56	F	BNT162b2	1	4	HZO	Oral acyclovir	Resolution
Thimmanagari et al. 2021 [23]	42	M	Johnson and Johnson	NA	7	HZO	Systemic and topical antivirus	Resolution
	49	M	Moderna	1	7	HZO	Systemic antivirus	Resolution
Li S et al. 2021 [20]	60	F	Sinovac	1	2	Recurrent herpes simplex keratitis	Topical ganciclovir	Resolution
Richardson et al. 2021 [24]	82	M	AstraZeneca	1	1	Recurrent herpes simplex keratitis	Oral acyclovir, topical ganciclovir, steroid, antibiotics	Resolution
Khan TA et al. 2021 [25]	48	M	COVISHIELD	2	21	Immune-Mediated Keratolysis	Oral acyclovir, topical steroid, antibiotics, anriglaucoma, tectonic penetrating keratoplasty	NA
Penbe et al. 2022 [26]	67	M	NA	NA	NA	PUK with nodular scleritis	oral steroids, azathioprine, topical cyclosporine, amniotic membrane grafting, penetrant keratoplasty	NA

SCI, subconjunctival injection; IV, intravenous; ICI, intracameral injection; HM, hand movement; NA, not mentioned in the article; LR-CLAL, living-related conjunctival limbal allograft; IVR, duration between vaccination and corneal events; HZO, herpes zoster ophthalmicus; PUK, peripheral ulcerative keratitis.

**Table 2 jcm-11-06828-t002:** Summary of corneal complications after COVID-19 vaccination.

Corneal Complications after COVID Vaccine		Vaccine					
		Moderna	BNT162b2	AstraZeneca/AZD1222	COVISHIELD	Johnson & Johnson	Sinovac
Acute corneal graft rejection	21 (56.8%)	7	8	2	3	0	1
Conjunctival limbal graft rejection	1 (2.7%)	1	0	0	0	0	0
HZO	11 (29.7%)	2	5	0	2	1	1
Herpes simplex keratitis	2 (5.4%)	0	0	1	0	0	1
Keratolysis	1 (2.7%)	0	0	0	1	0	0
PUK	1 (2.7%)	NA	NA	NA	NA	NA	NA
Total corneal AE	37	10 (27.0%)	13 (35.1%)	3 (8.1%)	6 (16.2%)	1 (2.7%)	3 (8.1%)

NA, not mentioned in the article; HZO, herpes zoster ophthalmicus; PUK, peripheral ulcerative keratitis; AE, adverse event.

**Table 3 jcm-11-06828-t003:** Summary of cornea transplant rejection events.

Cornea Transplant Rejection		
Age	Average	66
	Median	68
Sex	Male	11 (52.6%)
	Female	10 (47.4%)
Type of transplant	PKP	13 (68.4%)
	DSAEK	3 (10.5%)
	DMEK	5 (21.1%)
Past cornea disease	Keratoconus	5 (26.3%)
	FECD	5 (21.1%)
	PBK	4 (21.1%)
	Infectious keratitis	2 (10.5%)
	Corneal scar	2 (10.5%)
	Others or not mentioned	3 (10.5%)
Previous graft failure		9 (47.4%)
Vaccine dosage	First	14 (67%)
	Second	7 (33%)
Time from vaccine to rejection	Average (day)	10
	Medium (day)	7
Time from transplant to rejection	Average (month)	64
	Medium (month)	24
Outcome	Recovery	16 (78.9%)
	Progression or no change	3 (15.8%)
Recovery time	Average (week)	3.7
	Medium (week)	3

PKP, penetrating keratoplasty; DSAEK, Descemet’s stripping automated endothelial keratoplasty; DMEK, Descemet’s membrane endothelial keratoplasty; FECD, Fuchs endothelial corneal dystrophy; PBK, pseudophakic bullous keratopathy.

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
