# Peer review of "Corneal Complications after COVID-19 Vaccination: A Systemic Review"

_jcm, 2022, doi:10.3390/jcm11226828_

Round 1

Reviewer 1 Report

Please expand the conclusion section by providing specific suggestions for the prevention of disease recurrence after vaccination(s). For example, do you suggest antiviral agents in known herpetic disease ?

Author Response

Response 1: Thank you so much for your suggestion. We had expand the conclusion section for providing specific suggestions for prevention disease recurrence after vaccinations.

We suggest administration or increase steroids before the vaccine administration for graft rejection preventions, especially in high-risk patients. Higher risk of rejection was noted at inflammatory causes of limbal stem cell deficiency, including Steven–Johnson syndrome, mucous membrane pemphigoid. Due to low incidence of herpetic keratitis and lack of significant evidence, we do not suggest the prophylactic antiviral therapy in patients with history of herpetic keratitis.

Please see page 9, para 4. Conclusion, line 260-263.

Reviewer 2 Report

The work is interesting and brings new knowledge in the given subject area. In the presented review 27 reports (37 patients, 39 eyes) were used.

The methodology of the study is unclear. It should be supplemented with a flow chart for article selection. In the keywords, the authors used only the word "cornea" from ophthalmic terms. In my opinion, also the word "eye" as very general but nevertheless able to contain desirable records that need to be analyzed. The description of the method raises concerns that not all articles were found. 

Author Response

Point 1: The methodology of the study is unclear. It should be supplemented with a flow chart for article selection.

Response 1: Thank you so much for your suggestion. We have included a flow chart for article selection. Please see page 2, para 2. Materials and Methods.

Point 2: In the keywords, the authors used only the word "cornea" from ophthalmic terms. In my opinion, also the word "eye" as very general but nevertheless able to contain desirable records that need to be analyzed. The description of the method raises concerns that not all articles were found. 

Response 2: Thank you so much for your suggestion. The purpose of this literature review is to overview of COVID-19 vaccine-associated corneal adverse events. Ocular adverse events after vaccination included eyelid swelling, acute macular neuroretinopathy, central serous choroiretinopathy, acute anterior uveitis, panuveitis, choroiditis, optic neuritis, ischemic optic neuropathy. Therefore, the search keyword “cornea” is sufficient for this literature review.

Round 2

Reviewer 2 Report

The authors addressed my comments and completed the manuscript.

I thank them and accept the paper for publication.